# Evaluation of Dynamic Soil-Pile-Structure Interactive Behavior in Dry Sand by 3D Numerical Simulation

**Sun Yong Kwon [1] and Mintaek Yoo [2],***

[1] Division of Public Infrastructure Assessment, Korea Environment Institute (KEI), Sejong 30147, Korea
[2] Railroad Structure Research Team, Korea Railroad Research Institute (KRRI), Uiwang 16105, Korea
\* Correspondence: thezes03@krri.re.kr; Tel.: +82-31-460-5490

**Abstract:** A 3D numerical model based on finite-difference approximation was formulated to predict the dynamic soil-pile-structure interaction (SPSI) in dry sand. A non-linear elastic, Mohr–Coulomb plastic soil-constitutive model was adopted for the proposed methodology with a hysteretic damping model which can simulate nonlinear behavior of soil and an interface model which can predict separation and slippage between soil and pile according to the external load condition. Simplified continuum model was used to properly simulate the semi-infinite boundary and improve analysis efficiency. The proposed numerical model was validated by comparison with experimental results performed by Yoo (2013). Thereafter, a parametric study was also carried out to investigate the complex dynamic behavior of pile foundation under varying conditions. It was demonstrated that inertial force induced by superstructure is dominant for dynamic SPSI in dry sand whereas the kinematic force induced by soil deformation is relatively insignificant. Pile peak bending moment occurs at 30% of the pile length when pile length is no longer than 5 T and at about 30% of 5 T (1.6 T) when the pile length is longer than 5 T. The pile head fixity governed the peak bending moment profile of pile and affected the dynamic responses of the system in conjunction with other factors, such as pile rigidity.

**Keywords:** numerical analysis; soil-pile-structure interaction; dynamic behavior; kinematic force; inertial force

## 1. Introduction

Many civil and plant projects are constructed on soft ground, such as offshore locations and reclaimed land. Therefore, a pile foundation system that can transmit superstructure loads to a lower stiff layer is widely used, rather than a shallow foundation system that requires sufficient sub-ground bearing capacity. The pile foundation performs a significant role not only in supporting axial loads induced by the superstructure, but also in maintaining the superstructure safely from lateral loads, which can be induced by several external factors. Presently, one of the primary factors that cause unexpected large lateral loads emerging in Korea is the earthquake. Korean society experienced successive earthquakes in the past 3 years, accordingly a public consensus which should have appropriate and precise insight on seismic behavior and design condition to reduce damage and casualties is being strengthened. Earthquake-induced lateral loads to the pile foundation can be divided into two kinds of components, which are inertial force induced by the superstructure and kinematic force induced by soil movement. These representative lateral forces in the dynamic soil-pile-structure interaction can cause critical damage to the pile through different complex mechanisms, so proper prediction of these mechanisms is very important.

Pak and Gobert (1991) [1], Zeng and Rajapakse (1999) [2], Eskandari et al.(2013) [3], Ahmadi and Eskandari (2013) [4], Eskandari et al. (2014) [5] carried out meaningful studies using analytical treatment

to investigate dynamic soil-structure interaction. However, target system of the analytical model was mainly focused on shallow foundations such as rigid disk. Most of the previous studies that tried to estimate the seismic performance of the pile foundation using numerical models adopted a simplified approach, such as the 1D Winkler method. Miwa et al. (2006) [6], Chang et al. (2007) [7], Tahghighi and Konaghi (2007) [8], Liyanapathriana and Polous (2010) [9], Rovithis et al. (2013) [10], and Anoyatis and Lemnitzer (2017) [11] applied a 1D simplified approach to simulate the seismic behavior of pile foundations. Although the simplified approach is more convenient and faster than other methods, it does not guarantee accurate and reliable results because of the many assumptions inevitably generated during the simplifying procedure. On the other hand, 2D or 3D continuum modeling is the most straightforward and reliable approach if an appropriate model is adopted, though it is a very complex and time-consuming process. Martin and Chen (2005) [12], Uzuoka et al. (2007) [13], Cheng and Jeremic (2009) [14], Comodromos et al. (2009) [15], Kim et al. (2012) [16], Hamayoon et al. (2016) [17], and Azizkandi et al. (2018) [18] tried to develop a 3D continuum model to estimate the dynamic pile behavior; however, most of the studies did not perform satisfactory verification of the results obtained from numerical model; only a partial verification of the 1g shaking table tests, which cannot reproduce soil-confining pressure in the field, was carried out. The results obtained from numerical simulation are not an exact solution, but an approximate solution. Therefore, it is essential to verify the approximate solution drawn by numerical models using proper experimental results, which can be regarded as the exact solution. At this time, the results applied as an exact solution should be drawn by experiment, such as a real-scale shaking table test, which properly reproduces pile behavior in the field. However, a real-scale shaking table test under earthquake loading is very difficult to carry out. Accordingly, it is necessary to estimate the applicability of the proposed numerical model using a dynamic centrifuge test, which can simulate field-confining pressure and whose reliability has already been verified in various researches.

In this study, dynamic behavior of soil-pile-structure system observed in a centrifuge test is simulated by a proposed 3D continuum model based on the finite-difference method. The proposed modeling method was implemented in FLAC3D, and the detailed modeling methodologies and various soil dynamic properties were determined not only by parametric studies for diverse important factors, but also by the suggestion of a advanced approach based on Kwon et al. (2013). [19]. Validation of the proposed numerical model was carried out comparing internal pile responses of two model pile cases obtained from centrifuge tests and numerical simulation after modeling the dynamic centrifuge test performed by Yoo et al. (2013) [20] using the proposed numerical model. Thereafter, the dynamic behavior of pile was thoroughly investigated by performing a parametric study using a verified numerical model. Furthermore, theoretical concepts that can be predicted intuitively or were proposed by previous research were re-established and suggestions were included based on newly observed behaviors.

## 2. Modeling Methodology and Estimation of Input Parameters

There are several approximating methods used in numerical modeling; the two most common are the finite-element and finite-difference methods. Both are widely utilized in many fields and have advantages and disadvantages. Compared with the finite-element method, the finite-difference method has an advantage in predicting and evaluating dynamic soil-pile-structure interactions under earthquakes because the explicit method commonly used in the finite-difference method is appropriate for solving the problems in nonlinear, high-strain, and physically unstable conditions. In addition, mixed discretization, which is widely applied in finite-difference method, is known as a more rational approach than reduced integration in the finite-element method for treating such problems (Itasca Consulting Group, 2006 [21]). In this study, numerical modeling of the dynamic soil-pile-structure interaction observed in centrifuge tests based on finite-difference approximation was performed. FLAC3D, which is a commonly used finite-difference code in geotechnical engineering, was applied. Detailed modeling methodologies and estimation methods of dynamic soil properties

were proposed and determined by repetitive analysis and parametric study based on the theoretical basis of soil dynamics.

## 2.1. Dynamic Properties of Soil

The Mohr–Coulomb elasto-plastic soil constitutive model was adopted. This is a representative constitutive model in geotechnical engineering and is widely used in various research works; however, the nonlinear behavior of soil under a strong earthquake cannot be properly simulated with it. For the appropriate simulation of the dynamic behavior of soil-pile-structure system, the nonlinearity of soil under strong earthquake motion should be reasonably modeled. From this perspective, the proper estimation of initial shear modulus and nonlinear variation of shear modulus with shear strain is essential for accurate simulation. In this study, the hysteretic damping model was applied to consider not only the energy dissipation but also the nonlinearity of the soil modulus. As a fitting equation for the hysteretic damping model, the default model that can simulate $G/G_{max} - \gamma$ curve by using two constants, $L_1$ and $L_2$ was adopted. In this model, the $G/G_{max} - \gamma$ curve can be represented by a cubic equation with zero slope at both low and high strain, as in Equation (1) (Itasca Consulting Group, 2006 [21]).

$$M_t = s^2(3 - 2s) - \frac{6s(1 - s)}{L_2 - L_1} \log_{10} e \tag{1}$$

where $M_t$ is the tangent modulus, $s = (L_2\text{-}L)/(L_2\text{-}L_1)$, $L = \log_{10}(\gamma)$, $L_1$ and $L_2$ are the extreme values of L, $\gamma$ is the shear strain, and e is the void ratio.

When applying the hysteretic damping model, $L_1$ and $L_2$ should be properly estimated according to the soil type and analysis conditions to simulate nonlinear reduction of $G/G_{max}$ accurately. $L_1$ and $L_2$ represent the degradation rate and starting point of degradation of $G/G_{max}$, respectively. In this study, $G/G_{max} - \gamma$ curves with various $L_1$, $L_2$ were investigated and optimized values for $L_1$, $L_2$ were determined by comparison with the $G/G_{max} - \gamma$ curve obtained by triaxial compression test (Kim et al., 2012 [16]) with Jumunjin sand, which is Korean standard sand. Figure 1 shows the results of calibration of the $G/G_{max} - \gamma$ curve for Jumunjin sand between the lab test and numerical analysis. As a result, the trace of $G/G_{max}$ obtained from the numerical analysis matched that obtained from lab test well when $L_1$, $L_2$ were determined as 0.5, −3.65, respectively.

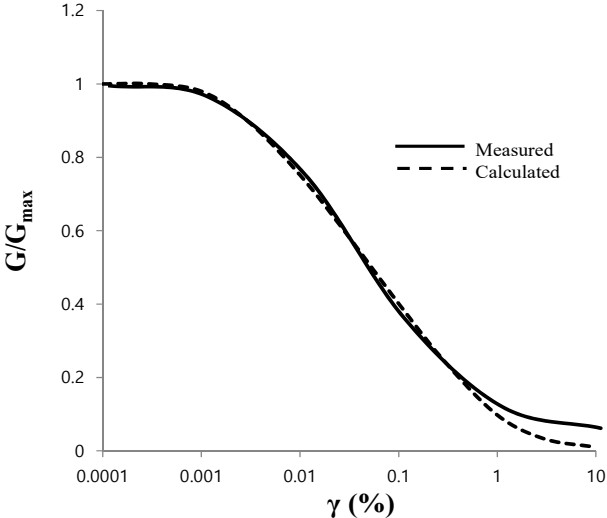

**Figure 1.** Comparison of measured and calculated $G/G_{max}$ curve.

The initial shear modulus generally depends on the soil-confining stress. In this study, the soil initial modulus was obtained by Hardin and Drnevich (1972) [22] as Equation (2). In the equation below, the values of empirical coefficients A, n were determined through regression analysis for the results of bender element tests which were performed by Yang (2009) [23]. In the regression analysis,

$G_{max}/F(e)P_a$ and $\sigma\prime_m/P_a$ for various test cases were plotted, and characteristic values such as A and n was induced.

$$G_{max} = AF(e)(OCR)^k P_a^{1-n}(\sigma\prime_m)^n \tag{2}$$

where $F(e) = \frac{1}{0.3+0.7e^2}$, e is the void ratio, $\sigma\prime_m$(kPa) is the mean principal effective stress ($\sigma\prime_m = (\sigma\prime_1 + \sigma\prime_2 + \sigma\prime_3)/3$), $P_a$(kPa) is the atmospheric pressure, k is the over-consolidation ratio exponent (see Table 1), and A and n are empirical coefficients determined as 247.73 and 0.567, respectively.

**Table 1.** Overconsolidation ratio exponent (Hardin and Drnevich, 1972).

| Plasticity Index | k |
|:---:|:---:|
| 0 | 0.00 |
| 20 | 0.18 |
| 40 | 0.30 |
| 60 | 0.41 |
| 80 | 0.48 |
| ≥100 | 0.50 |

### 2.2. Soil-Pile Interface Model

Three types of situations, namely full contact, slippage, and separation, could occur between the soil and pile surface under strong earthquake loading. The soil-pile interface element should be able to reproduce these kinds of phenomena to predict the dynamic soil-pile interactive behavior accurately. In this study, the interface element that can consider both slippage and separation between the soil and pile under strong earthquake conditions was adopted. The applied interface model estimates the soil stiffness using normal and shear spring constants, which were calculated from Equation (3) (Itastca Consulting Group, 2006 [21]). Soil nonlinearity was considered in the shear modulus (G) and bulk modulus (K), which were input to Equation (3). The interface stiffness obtained by Equation (3) was input continuously according to the depth by FISH. On the other hand, the interface yield criteria were set according to Equation (4) and three types of interface behaviors such as fully contact, slippage, and separation occurred with the magnitude of external force ($F_n$).

$$k_E = max\left[\frac{K + (4/3)G}{\Delta z_{min}}\right] \tag{3}$$

where K is the bulk modulus, G is the shear modulus, and $\Delta z_{min}$ is the smallest width of the adjacent zone in the normal direction

$$F_{smax} = cA + tan\varnothing(F_n - pA) \tag{4}$$

where c is the cohesion along the interface, $\varnothing$ is the friction angle of the interface, p is the pore water pressure, and A is representative area associated with the interface node.

The soil at soil-pile interface zone has different strength characteristic with that in remote zone because of the soil-pile interaction and soil disturbance. Therefore, in this study, interface friction angle which is slightly smaller than maximum internal friction angle of the remote soil that is not affected by soil-pile interaction was adopted. Kraft (1990) [24] proposed a value that is approximately 70% of the maximum internal friction angle of the remote soil as the calculation criterion for the internal friction angle at the interface. Reddy et al. (2000) [25] proposed a value that is approximately 60% of the maximum internal friction angle of the remote soil. In this study, using the results obtained by Randolph et al. (1994) [26], the internal friction angle at the soil–pile interface element was calculated using Equation (5).

$$\delta = \varnothing_{max} - 5^\circ \tag{5}$$

where δ is the internal friction angle of the interface, and $\varnothing_{max}$ is the maximum internal friction angle of the remote soil.

### 2.3. Modeling of Far-Field Boundary

One of the most important points in dynamic analysis for a soil-pile system is proper simulation of semi-infinite boundary. If soil elements are generated without limit for accurate simulation of the infinite boundary in a field, the effectiveness of the analysis is significantly decreased by the long analysis time. Therefore, setting up the boundary condition at a proper distance is very important. It must be considered carefully because inaccurate results can be induced due to reflected waves that are generated at the model boundary when the boundary condition is not properly modeled. In this study, a simplified continuum modeling method, which was proposed by Kim et al. (2012) [16], was adopted to simulate the soil-boundary responses accurately and effectively.

Figure 2 depicts the 3D schematic view of the model mesh used in this study. The entire region is divided into two parts: near-field and far-field zones. As shown in this figure, only the near field, which is directly affected by the soil-pile dynamic interaction, was physically modeled, whereas the far field, which is considered to be unaffected by the soil-pile dynamic interaction, was not modeled. The acceleration–time histories along depth in far-field zone were then obtained independently by preliminary site-response analysis and input at the outer boundary of near-field zone. This kind of boundary is advantages in that it can reduce the analysis time remarkably and is very effective for the simulation of the boundary for a semi-infinite soil condition compared with the full modeling of the two zones. Nogami et al. (1992) [27], Wang et al. (1998) [28], Boulanger et al. (1999) [29], and Assareh & Asgarian (2008) [30] carried out similar approaches, relating it with p–y elements; however, most of those were performed with one- or two- dimensional simplified approaches. Application of this method to the 3D continuum model using the finite-element or finite-difference method is needed to reach a straightforward analysis of the prototype system. Kim et al. (2012) [16] applied this kind of method to finite-difference analysis, and the applied model was calibrated by 1g shaking table tests, which cannot reproduce the confining pressure of field conditions.

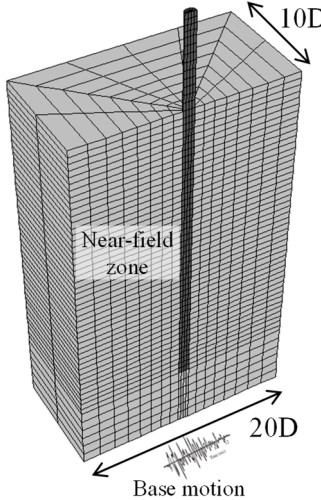

**Figure 2.** Three-dimensional schematic view of model mesh.

In this method, the boundary extent of the near-field zone, where the soil motion becomes the same as the far-field motion, should be determined. The acceleration amplification ratio, which is the ratio of the maximum acceleration at the surface to the base input acceleration, was analyzed to determine the optimum boundary extent of the near-field zone. Figure 3 shows the amplification ratio with respect to the distance from the pile center. The dynamic excitation of the pile increases the surface acceleration; hence, the ratio decreases as the distance from the pile is increased. The optimal distance, at which the ratio is constant, was determined as 10D, where D is the pile diameter. Finally, the boundary applied as the extent of the near-field zone was determined as 10D from the pile center.

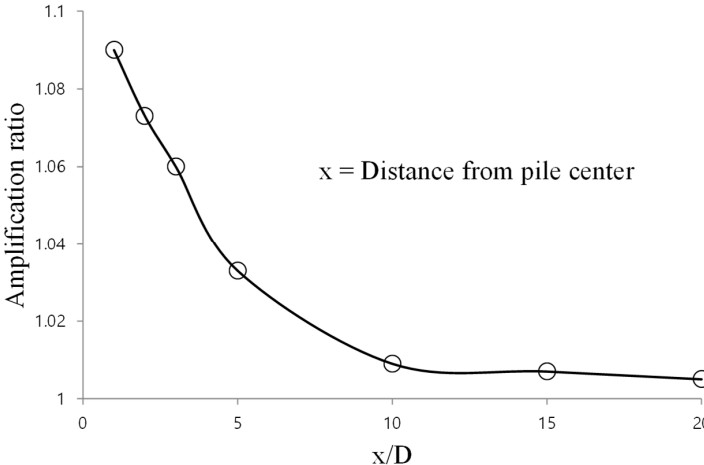

**Figure 3.** Acceleration-amplification ratio of soil according to the distance from pile.

*2.4. Modeling Details and Analysis Procedure*

For effectiveness, the axisymmetric model of Figure 2 was used and earthquake motion was induced in the x direction. Each earthquake motion was input at the bottom of the model in the form of an acceleration–time profile for rigid-base input motion boundary condition. Pile foundation was modeled by elastic with cylinder-type solid element. The element size of the numerical model was set satisfying Equation (6) for the maximum frequency content of input motion (Kuhlemeyer and Lysmer, 1973) [31]. As a result, the pile was modeled by a radial cylinder-shaped solid element with about 46 elements in the vertical direction, and its material was modeled applying the elastic model. It was confirmed that the pile stress induced during analysis was much lower than the material yielding stress.

$$\Delta Z \leq \frac{V_s}{10f} \tag{6}$$

where f is the maximum input frequency, $V_s$ is the shear wave velocity, and $\Delta Z$ is the element size.

The numerical analyses progressed following three steps: (1) simulation of geostatic equilibrium, (2) pile installation, and (3) application of dynamic loading. In the first step, the in-situ geostatic condition was achieved to satisfy stress equilibrium in soil under gravitational load. In the second step, soil elements corresponding to the pile zone were removed and the pile elements were generated with the soil-pile-interface elements. The stress equilibrium condition was again achieved against the stress increase due to pile material weight. Finally, dynamic loading, in the form of the acceleration–time history of the input earthquake motion, was applied to the nodes on the bottom boundary, and the time histories of the horizontal acceleration in the far-field zone were applied to the corresponding nodes of the left and right boundaries. Superstructure weight on pile head was properly simulated by adjusting unit weight of the corresponding elements.

## 3. Validation of Proposed Model

To improve the reliability and minimize the error of the proposed numerical model, a validation procedure was carried out by comparing the dynamic pile internal responses calculated by the numerical model with those measured by the dynamic centrifuge model tests (Yoo et al., 2013) [20]. The centrifuge model tests were performed with the KAIST dynamic centrifuge facility in Korea, which has a 5 m radius, 2.5 ton payload and a up to 100 g centrifugal acceleration. All tests were performed at a centrifugal acceleration of 40 g and similitude factor was 40, also. In order to analyze dynamic behavior of soil-pile interaction using centrifuge tests, the model structure should be manufactured considering reasonable similitude law. The model piles and ground were simulated by dynamic centrifuge similitude law which was verified by many preceded researches (Schofield et al. (1980) [32], Taylor (1995) [33]). Two aluminum model pile were used with 2.5 cm (1 m in prototype scale) and

1.8 cm (0.72 m in prototype scale) external diameters and a 0.1 cm (0.04 m in prototype scale) thickness, and embedded depth was 57 cm (22.8 m in prototype scale). Joomunjin sand, characterized as clean and uniform sand, was used in these tests. The primary properties of model soil and pile are summarized in Tables 2 and 3. The layout of the test is shown in Figure 4. Eight pairs of strain gauges were attached on both sides of the pile to calculate the bending moment in the pile during vibration. Eight accelerometers were installed in the soil at the same depth as each strain gauge to calculate the displacement of soil and one accelerometer was attached to the pile head to measure the acceleration responses of the pile. In order to compare centrifuge model tests and numerical analysis results, all results from the centrifuge tests were expressed at the prototype scale.

**Table 2.** Input properties of model soil.

| Property | Value |
|---|---|
| Friction angle (degree) | 42 |
| Dry density (kN/m$^3$) | 15.80 |
| Poisson's ratio | 0.3 |
| Void ratio | 0.677 |
| Relative density (%) | 80 |

**Table 3.** Properties of the model piles.

| Property | Model 1 | Model 2 |
|---|---|---|
| Scaling relation | 40 | 40 |
| Diameter of pile (m) | 0.025 (1.00*) | 0.018 (0.72*) |
| Thickness of pile (m) | 0.001 (0.04*) | 0.001 (0.04*) |
| Flexural rigidity (N · m$^2$) | 376083 (9.63E + 11*) | 133889 (3.43E + 11*) |
| Embedment depth (m) | 0.57 (22.8*) | 0.57 (22.8*) |
| Concentrated mass (kg) | 1.4 (89600*) | 1.4 (89600*) |

\* Prototype scale.

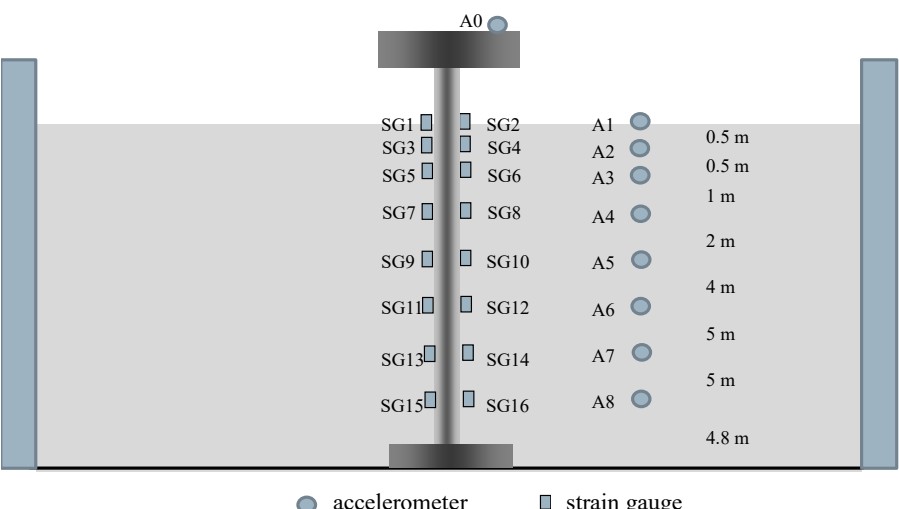

**Figure 4.** Layout of test (in prototype).

Figure 5a depicts an example of the acceleration–time history of a sinusoidal event with a maximum acceleration of 0.4 g and frequency of 1 Hz. Two kinds of real earthquake events (Ofunato and Nisqually earthquakes) were scaled into various input motions with different amplitudes of peak base accelerations (0.06 g, 0.13 g, 0.25 g, 0.36 g, 0.51 g). Figure 5b,c depict the representative acceleration–time histories of those two real-earthquake events, which have a maximum amplitude of 0.51 g.

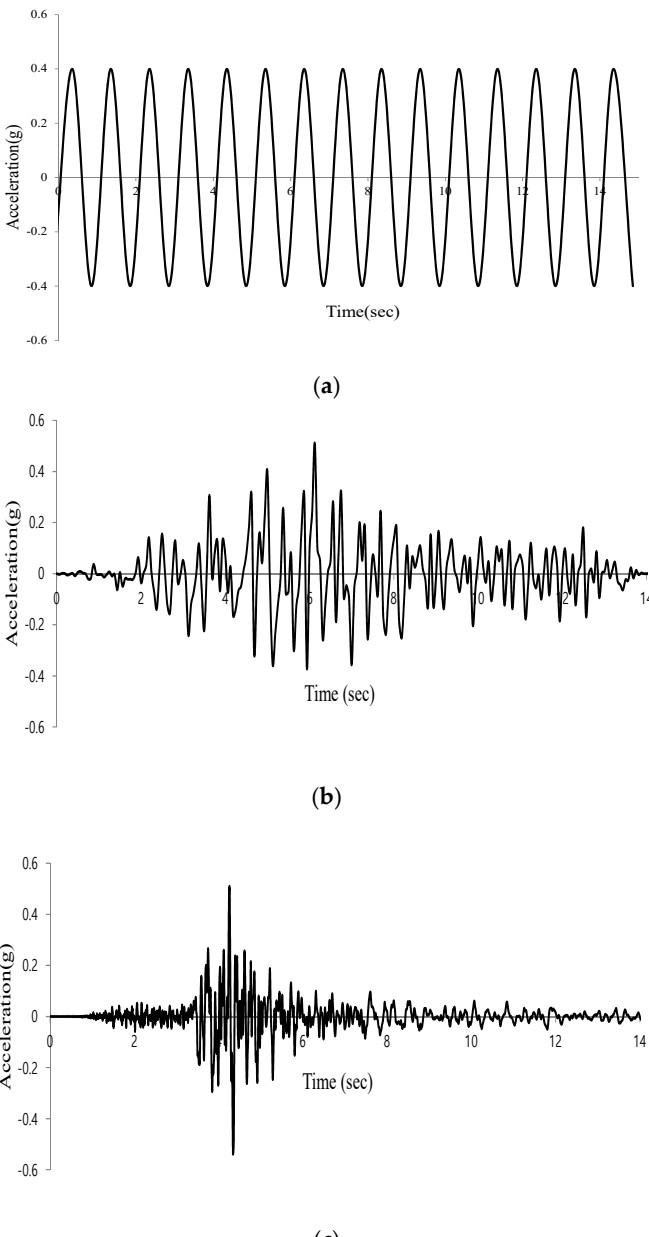

**Figure 5.** Acceleration–time history of input motion: (**a**) Acceleration-time history of base input for sine wave (1 Hz, 0.4 g); (**b**) Acceleration-time history of base input for Ofunato earthquake ($a_{max}$ = 0.51 g); (**c**) Acceleration-time history of base input for Nisqually earthquake ($a_{max}$ = 0.51 g).

For the first stage of the validation procedure, the internal pile responses calculated from the numerical model and those measured from centrifuge tests using the test case of Model 1 in Table 3 were compared and discussed. Figure 6a,b shows measured and calculated peak bending moment profiles for different input frequencies under input acceleration of 0.13 g. It was identified that the peak bending-moment profiles predicted by the proposed numerical method showed similar trend for various input frequencies with those observed in centrifuge tests. The peak bending moment of the pile obtained in both approaches sharply increased as input frequency approaches to 1 Hz. Figure 7a–c shows the maximum values of the pile bending moments for various input earthquake frequencies and amplitudes. As shown in Figure 7, the maximum values calculated by numerical model and measured by the centrifuge test show good agreement for various input earthquake conditions. Based on Figures 6 and 7, it can be identified that resonance occurred in both results when the frequency

of the input earthquake approached 1 Hz, meaning that the natural period of the system appeared to about 1 s in both results. This kind of behavior was also captured in sweep analysis as shown in Figure 8. Therefore, it is demonstrated that proposed numerical model can simulate the important dynamic characteristics of the soil-pile-structure system, such as resonance.

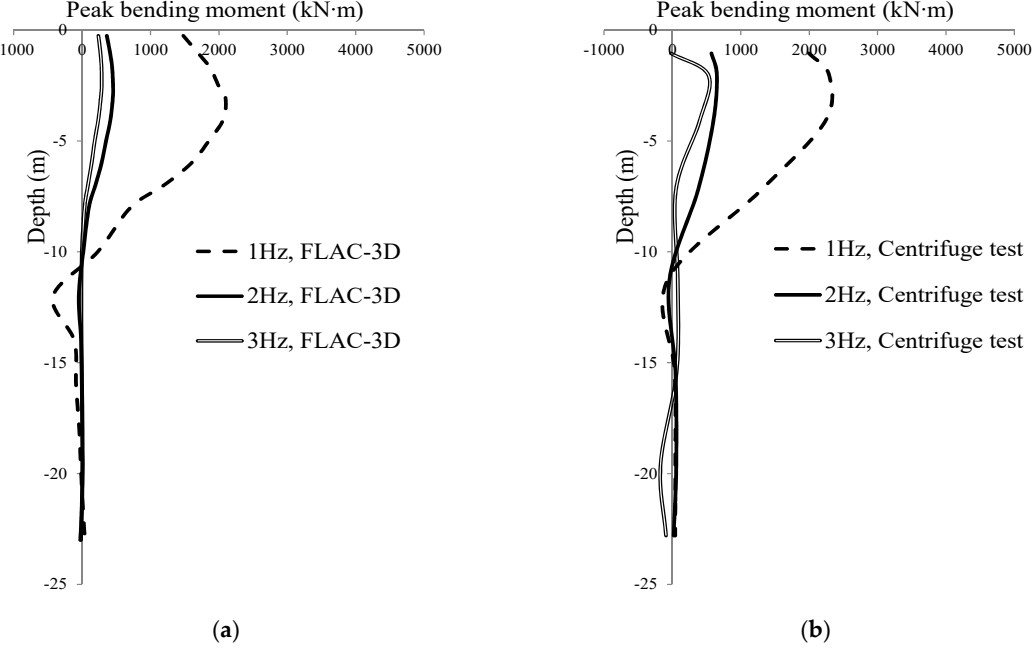

Figure 6. Measured and calculated peak bending moment profiles under input acceleration of 0.13 g (Model 1, sinusoidal wave): (**a**) Calculated by proposed model; (**b**) Measured by experiment.

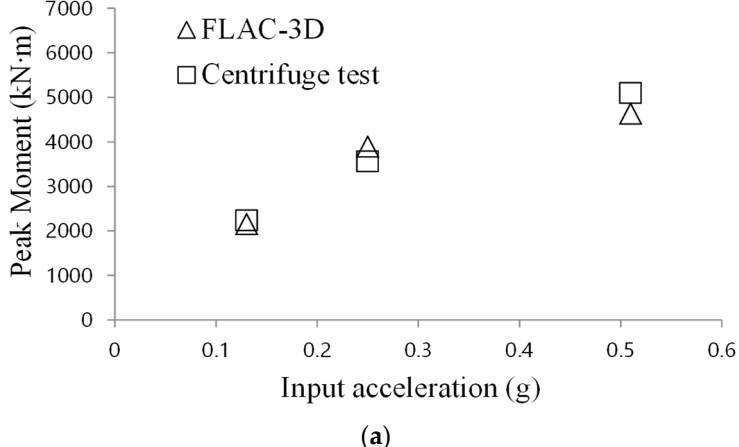

(**a**)

**Figure 7.** *Cont.*

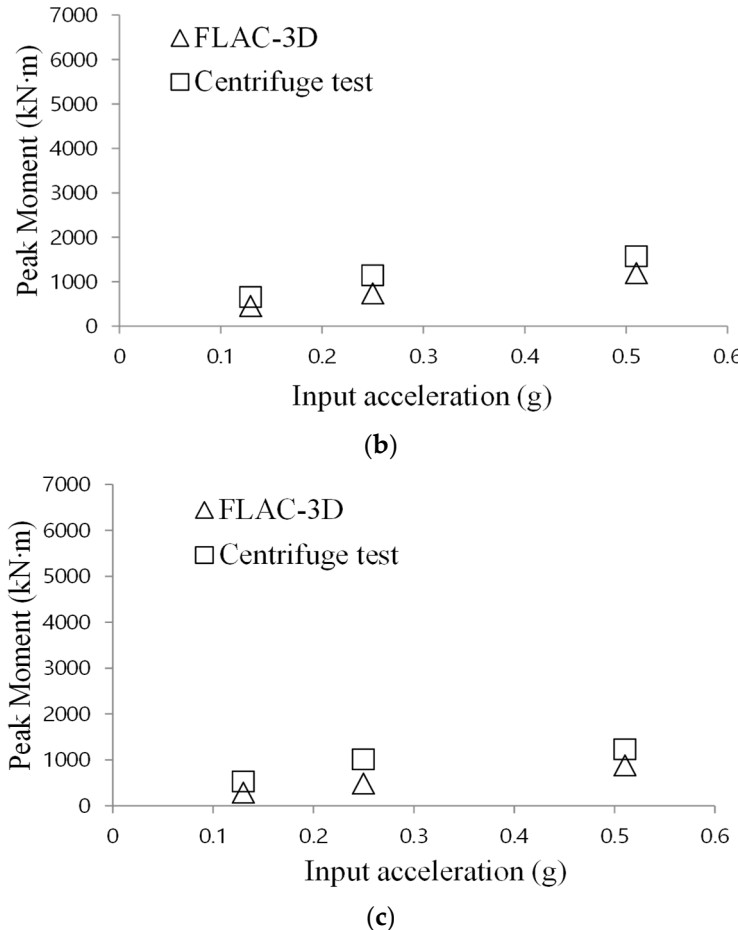

**Figure 7.** Maximum values of bending moments for various input motions (Model 1, sinusoidal wave): (**a**) Input frequency:1 Hz; (**b**) Input frequency:2 Hz; (**c**) Input frequency:3 Hz.

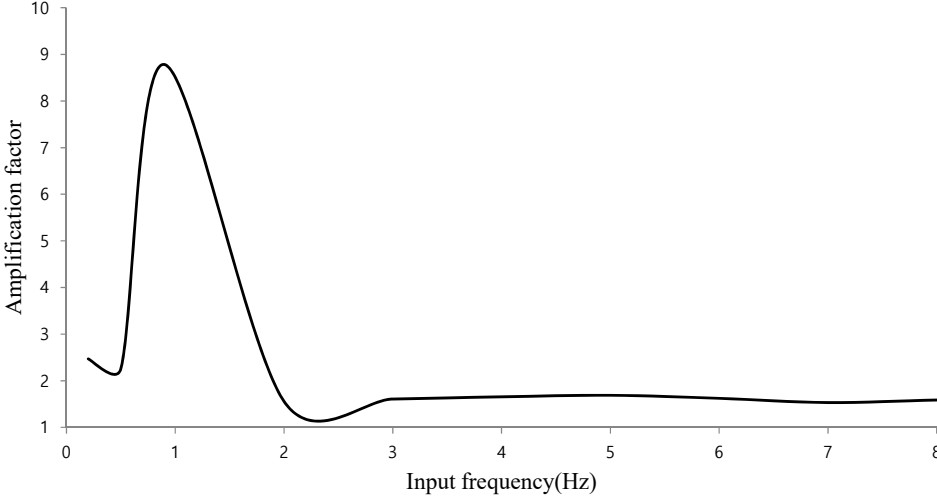

**Figure 8.** Result of sweep analysis.

Figures 9 and 10 show the peak bending moment profiles for different input accelerations when the Nisqually and Ofunato earthquakes were applied as input excitations, respectively. The peak bending moment according to the depth predicted by the numerical model matched well with those observed in dynamic centrifuge tests for both real earthquake events. It is shown that the proposed

numerical model has the capability to properly simulate seismic pile behavior for various input earthquake conditions.

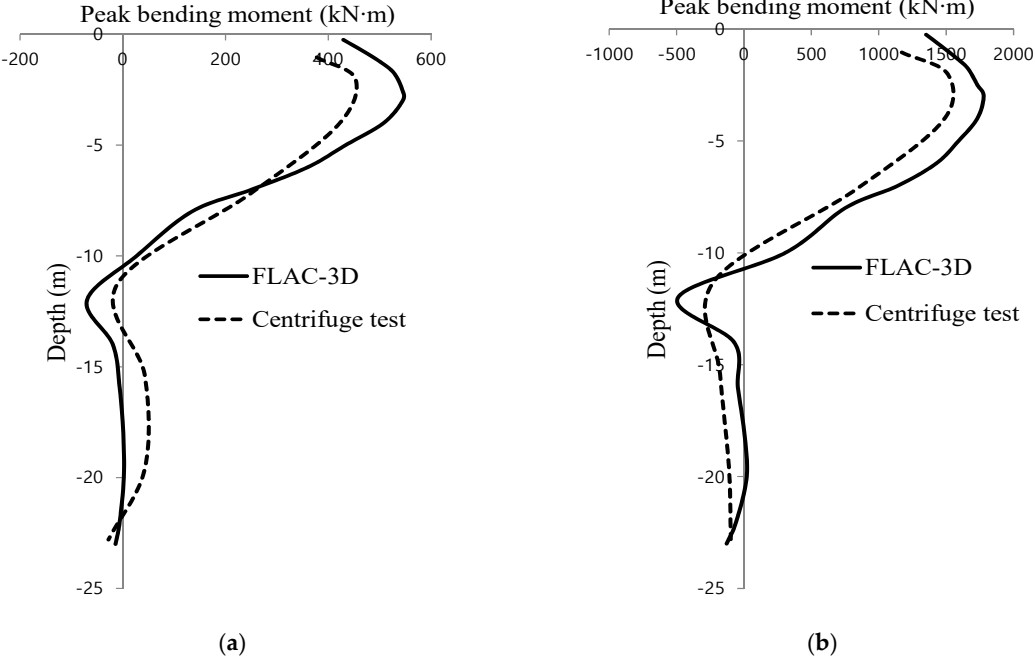

(**a**)                                    (**b**)

**Figure 9.** Measured and calculated peak bending moment profiles (Model 1, Nisqually earthquake): (**a**) Nisqually, 0.13 g; (**b**) Nisqually, 0.51 g.

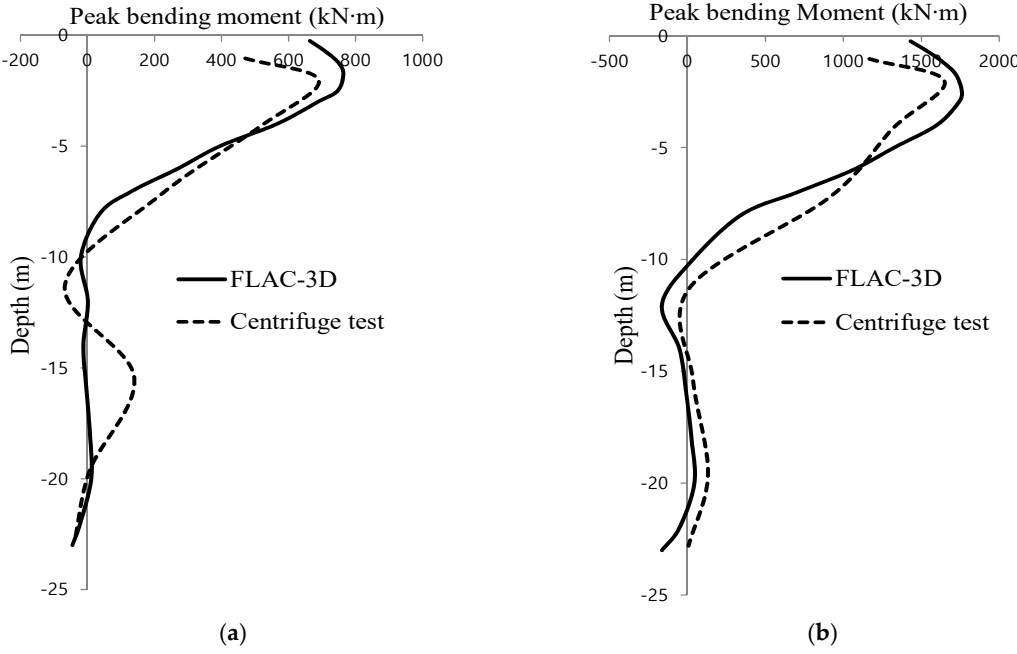

(**a**)                                    (**b**)

**Figure 10.** Measured and calculated peak bending moment profiles (Model 1, Ofunato earthquake): (**a**) Ofunato, 0.13 g; (**b**) Ofunato, 0.51 g.

Figure 11a,b shows the comparison between the analysed and observed maximum values of pile bending moments for various input amplitudes of both real-earthquake events. And, in this figure, discrepancies between two results were also showed for quantitative discussion. It depicts that all results show good agreement with discrepancy rate of less than 20%, which indicates that the predicted dynamic-pile responses agree reasonably well with the observed values for both the Nisqually and

Ofunato earthquakes. In particular, the numerical model predicted the pile bending moments well in conservative manner with good accuracy, which means that the proposed numerical model can simulate the dynamic pile behavior in dry sand with safety and economic margins under various kinds of earthquake conditions.

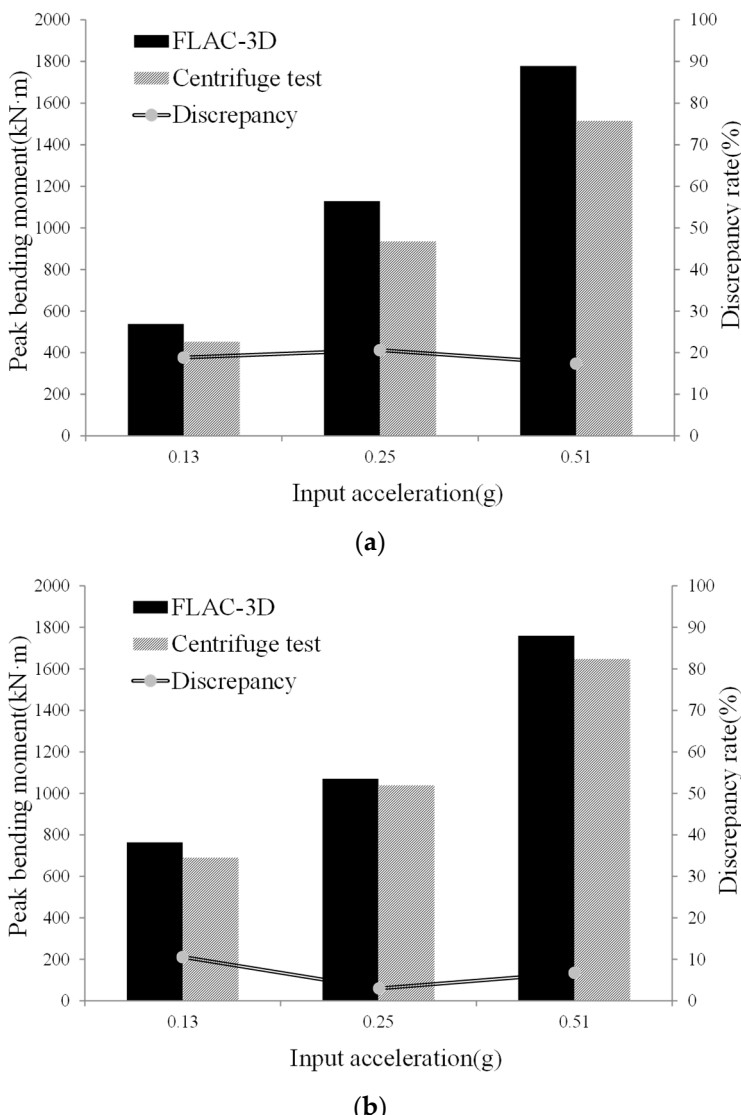

**Figure 11.** Measured and calculated maximum values of bending moments for various input amplitudes (Model 1): (**a**) Nisqually earthquake; (**b**) Ofunato earthquake.

The above results identified the viability of the proposed model for simulating the dynamic behavior of piles under earthquake conditions. However, the validation was performed only for a single test pile case, which was carried out using Model 1. Additional verification is necessary to validate the applicability of the proposed model for various site conditions.

For this reason, further verification was conducted by comparing the internal pile responses of Model 2. In this part, only the test case in which the Nisqually earthquake was applied as an input excitation was used because test results obtained from the test case applying Ofunato earthquake as an input excitation were unreasonable due to defects in the measuring instruments during the test. Figure 12 shows the comparison between the observed and predicted maximum values of pile bending moments for various input amplitudes in a single graph. It can be seen that the calculated bending

moments agree quite well with the values recorded during the experiment. Discrepancies for every case were less than 10%.

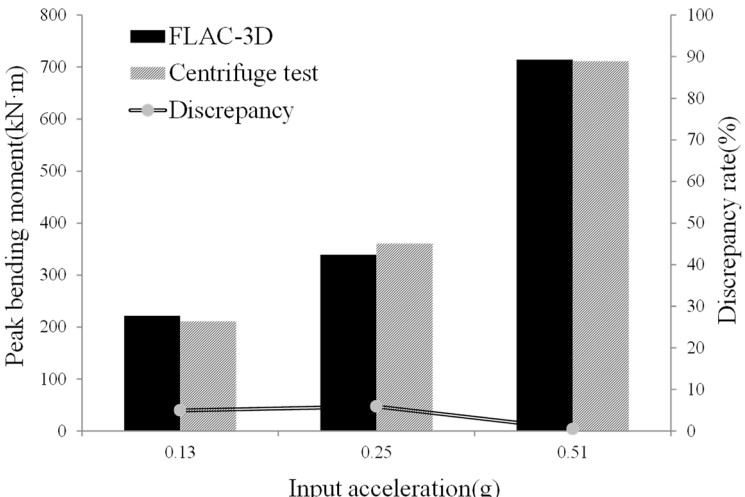

**Figure 12.** Measured and calculated maximum values of bending moments for various input amplitudes for Nisqually earthquake (Model 2, Nisqually earthquake).

In accordance with a series of verifications, it is confirmed that the proposed model has the capability to simulate the dynamic soil-pile-structure interaction in dry sand and to predict seismic behavior of the pile foundations for various site conditions.

## 4. Parametric Study

Numerical simulation is very effective because, if the numerical model is properly established, many different cases of the target problem can be analyzed simultaneously with slight modification of the input properties. The condition in which a dynamic external load is exerted to the soil-pile-structure system is much more complex than the condition in which a static load is exerted because far more factors affect the behavior of the system in the dynamic condition. Every single factor sensitively affects the seismic performance of the entire system and the extent of this effect is significantly different according to the conditions of the system. Therefore, the effect of the primary factors that govern the dynamic behavior of the soil-pile-structure system must be investigated for a dependable seismic design of the pile foundation.

In this section, the results of parametric studies are presented to investigate the seismic behavior of pile foundations under various conditions. The applicability of the proposed model was verified once again by comparing the results of the model with those from previous research. Parametric studies were carried out by varying the weight of the superstructure, soil relative density, pile length, and pile head fixity to estimate effect of each parameter on the dynamic behavior of pile foundation. The input properties are fundamentally identical to those of the previous section and the detailed properties are determined in accordance with relative density condition. The determined input properties for different relative densities of 30%, 50%, and 80% are listed in Table 4. The properties for different relative densities were determined by Kumar and Madhusudhan (2010) [34]. Sine waves were applied as the input excitation to investigate the dynamic behavior of the soil-pile system under various input earthquake conditions.

**Table 4.** Input properties of model soil for parametric study.

| Property | Values for $D_r$ = 35 % | Values for $D_r$ = 55 % | Values for $D_r$ = 80 % |
|---|---|---|---|
| Friction angle (°) | 39 | 40.5 | 42 |
| Dry density (kN/m$^3$) | 14.2 | 14.8 | 15.8 |
| Poisson's ratio | 0.32 | 0.31 | 0.30 |
| Void ratio | 0.851 | 0.782 | 0.677 |

### 4.1. Effect of Superstructure Weight on Dynamic Behaviour of Pile

Inertial force induced by superstructure and kinematic force induced by soil movement is representative in dynamic soil-pile-structure interaction. Therefore, in this section, influence of these two representative dynamic forces on model pile used in this study was investigated. For the first step, numerical analyses using proposed numerical model were performed for four systems which have different weight of superstructure. Figure 13a,b depicts maximum lateral displacement profiles of pile for various input accelerations and four different weights of superstructure, which was 0 kN, 300 kN, 600 kN, 900 kN. Resonance effect was excluded in this section because natural frequency of each system was about 1.5~3 Hz according to the sweep analysis. It is identified that weight of superstructure significantly affected on dynamic lateral responses of pile. When weight of superstructure decreases from 900 kN to 0 kN, maximum lateral displacement of pile decreases about 90%. This kind of behavior was observed regardless of input acceleration level. Because test condition was identical in all cases except weight of superstructure, this phenomenon is responsible for difference of inertial force induced by superstructure and exerted kinematic force is identical for all analysis cases. It can be noted that inertial force induced by superstructure significantly governs seismic behavior of soil-pile-structure system in dry sand.

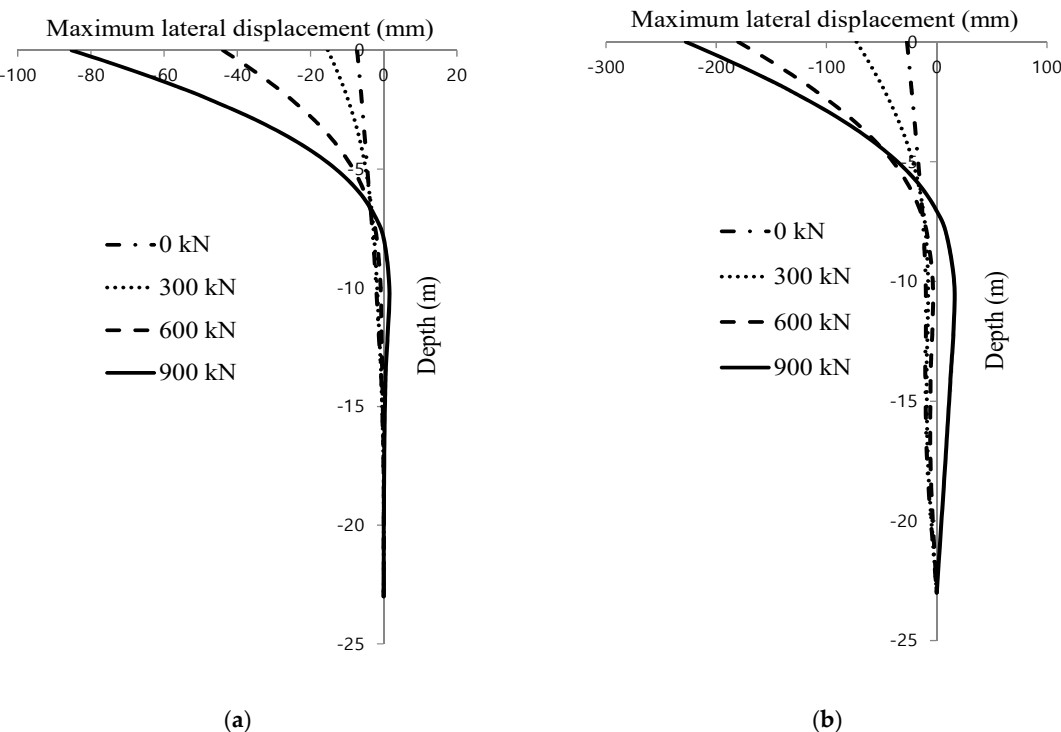

(**a**)　　　　　　　　　　　(**b**)

**Figure 13.** Maximum lateral pile displacement envelopes obtained from four different weight of superstucture. (**a**) Input acceleration: 0.13 g; (**b**) Input acceleration: 0.45 g.

Figure 14 depicts variation of maximum lateral displacement of pile for different input acceleration. As weight of the superstructure increases and input acceleration increases, maximum value of lateral pile displacement also increases. Additionally, relationship between weight of superstructure and maximum value of lateral pile displacement shows linear with high correlation coefficient for various input acceleration conditions. It can be demonstrated that lateral displacement of pile under specific axial load can be properly predicted if proposed model is adopted in practice. Finally, critical damage of entire structure as well as pile foundation can be prevented by estimating dynamic lateral responses of pile foundation under any earthquake level by proposed numerical model.

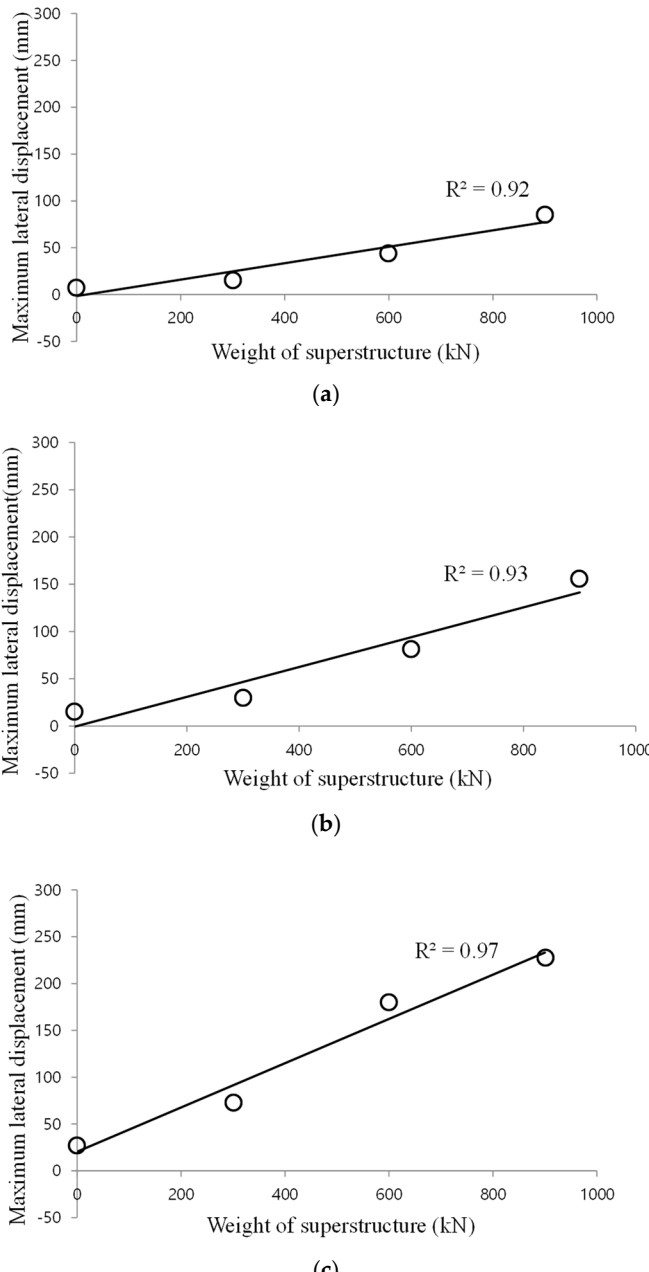

**Figure 14.** Relationship between weight of superstructure and maximum lateral pile displacement. (**a**) Input acceleration: 0.13 g; (**b**) Input acceleration: 0.25 g; (**c**) Input acceleration: 0.45 g.

### 4.2. Effect of Relative Density on Dynamic Behaviour of Pile

Repeated analyses were carried out to investigate effect of soil relative density on the dynamic behavior of pile for various soil relative densities of 30%, 50%, and 80%. Figure 15 depicts the peak bending moment profiles of three systems for two different input accelerations. As shown in the figure, the bending moment profiles are almost identical for all cases, although the soil relative density is significantly different in each case. Figure 16 depicts the maximum lateral displacement profiles of the pile. It shows a similar pattern to that of the peak bending-moment profiles. This is quite a different trend from the previous result of the weight of the superstructure. In this part, the analysis condition was identical for all cases except soil relative density. Therefore, the induced inertial force was constant in every system and only the kinematic force induced by soil deformation was different. This means that the kinematic force is relatively not significant in dry sand.

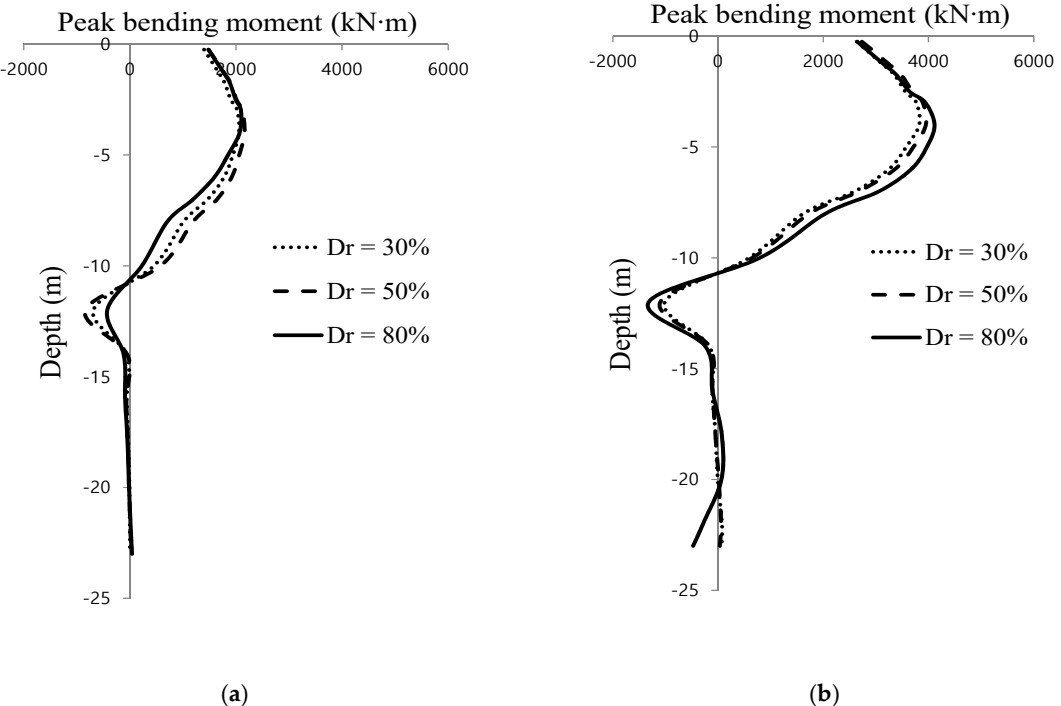

(**a**)  (**b**)

**Figure 15.** Peak bending moment envelopes obtained from three different relative densities. (**a**) Input acceleration: 0.13 g; (**b**) Input acceleration: 0.25 g.

Several meaningful characteristics of dynamic soil-pile-structure interaction in dry sand were estimated through a series of numerical analyses, which were described in this section. In dry sand, the effect of the inertial force induced by the superstructure is dominant, whereas the effect of the kinematic force induced by soil movement is not as significant. This kind of dynamic characteristic must be properly integrated when the seismic design of the pile foundation embedded in dry sand is determined.

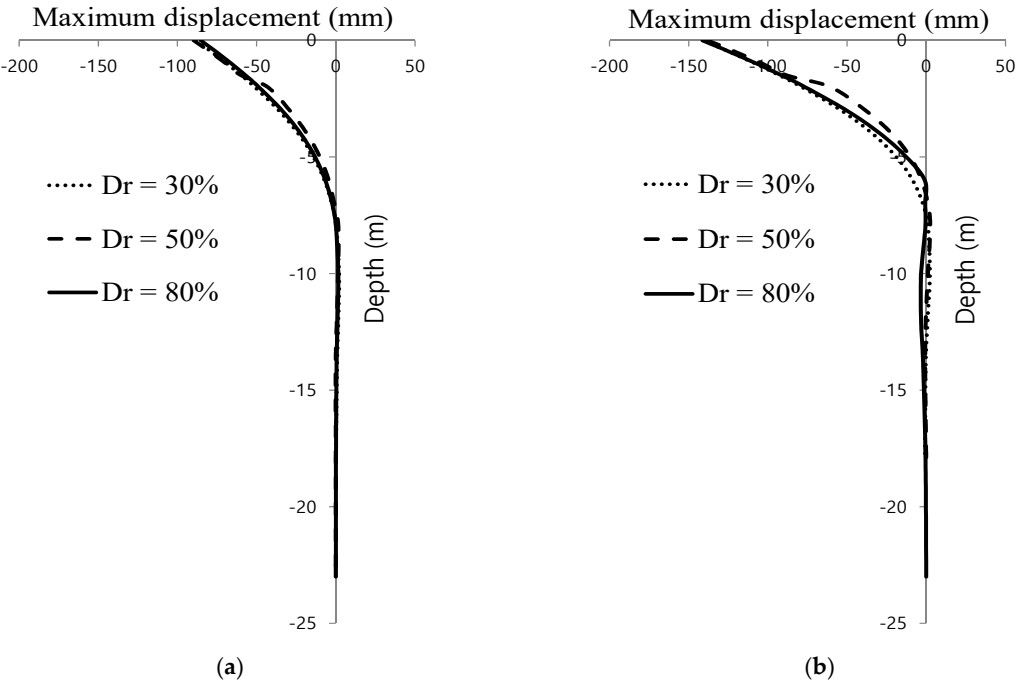

**Figure 16.** Maximum lateral pile displacement envelopes obtained from three different relative densities. (**a**) Input acceleration: 0.13 g; (**b**) Input acceleration: 0.25 g.

### 4.3. Effect of Pile Length on Dynamic Behaviour of Pile

Another repetitive analyses were carried out for 9 model systems with different pile lengths of 2 m (1.1 T), 4 m (2.2 T), 5 m (2.7 T), 8 m (4.4T ), 9 m (4.9 T), 10 m (5.5 T), 12 m (6.6 T), 18 m (9.8 T), and 23 m (12.6 T) to examine the effect of pile length on the dynamic behavior of the pile. Each pile length was marked as the characteristic length of pile (T) which was defined by the analytical approach performed by Matlock and Reese (1960) [35] as Equation (7).

$$T = \sqrt[5]{\frac{EI}{n_h}} \tag{7}$$

where $n_h$ is the soil reaction constant and EI is flexural rigidity of pile.

Figure 17 depicts the comparison of the maximum lateral pile displacement profiles. When the pile length exceeds 5 T, the maximum lateral displacement profile shows long-pile behavior. In contrast, when the pile length is shortened to below 3 T, the maximum lateral displacement profile shows short-pile behavior which has no negative bending moment. When the pile length is between 3 T and 5 T, the maximum lateral displacement profile shows intermediate behavior. Figure 18 shows the relationship between pile length and rotational depth, where each parameter is normalized by T. In this figure, rotational depth means the distance between the ground surface and the rotation axis of the pile. It is identified that each parameter shows a linear relation with a high correlation coefficient. This means that the pile rotates at a depth of about 70% of the pile length when the pile behaves as a short pile.

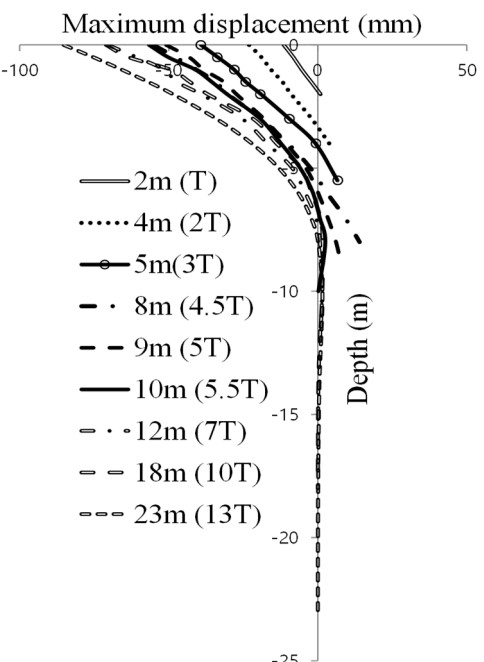

**Figure 17.** Maximum lateral pile displacements for various pile lengths (input acceleration: 0.13 g).

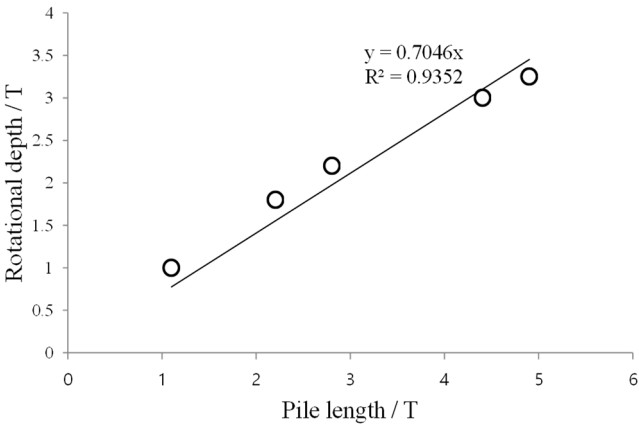

**Figure 18.** Relationship between pile length and rotational depth.

Figure 19 depicts a comparison of the peak bending-moment profile. The trend in the bending moment was very similar to that of the lateral pile displacement. When the pile length exceeds 5 T, the bending-moment profile shows long-pile behavior. When the pile length is less than 5 T, the bending moment profile of a short pile was appeared. Figure 20 shows the occurrence depth of the peak bending moment according to the pile length. It is identified that the occurrence depth of the peak bending moment increases as the pile length increases when the pile length is shorter than 5 T. This kind of behavior appears when the pile length is no longer than 5 T, and the occurrence depth of the peak bending moment is about 30% of the pile length. However, if the pile length becomes longer than 5 T, the occurrence depth of the peak bending moment no longer increases, but rather is converged. Based on this phenomenon, it can be suggested that the peak bending moment occurs at 30% of the pile length when pile length is no longer than 5 T and at about 30% of 5 T (1.6 T) when the pile length is longer than 5 T. Through the obtained dynamic responses of the pile for different pile lengths, it is demonstrated that dynamic pile responses are significantly different for pile length. It strongly affected not only in response trend according to the depth but also in rotational depth and occurrence depth of the peak bending moment. And it is identified that proposed numerical model can

capture complex dynamic characteristics of soil-pile-structure interactive behavior, also can provide some important point of view on practical seismic design of pile foundation.

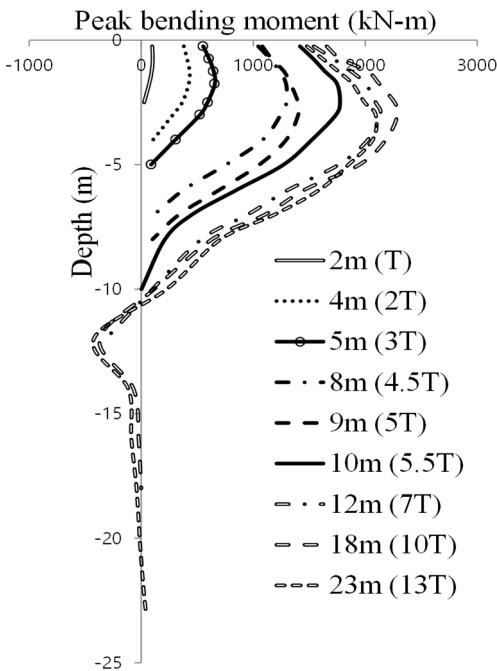

**Figure 19.** Peak bending moment profiles for various pile lengths.

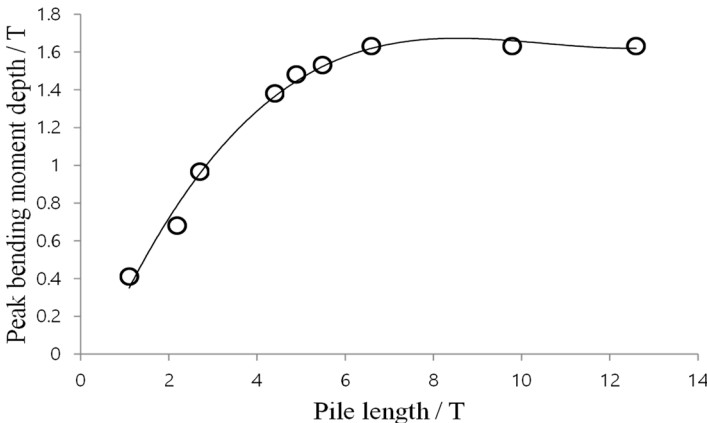

**Figure 20.** Relationship between pile length and peak bending moment depth.

### 4.4. Effect of Pile Head Fixity on Dynamic Behaviour of Pile

Two kinds of pile head boundary conditions were considered to investigate effect of pile head fixity on the dynamic behavior of piles. The pile head fixity was controlled to be fixed or free in FLAC3D by adjusting the constrain command at the level of the superstructure. Figure 21 depicts the comparison of the peak bending-moment profiles for piles with two different lengths of 23 m (13 T), 8 m (4.5 T). In this figure, it is shown that the maximum value of the pile bending moment is induced at different depths with different pile head fixities. For the free-head system, the peak bending moment occurred at a certain depth below the ground surface regardless of pile length. For the fixed-head system, the peak bending moment consistently occurred at top of the pile. On the other hand, in Figure 21b, the peak bending moment observed from the fixed-head system was significantly lower than that of Figure 21a. This is because the pile length of Figure 21b is relatively short (4.4 T), which is in the intermediate length range. Therefore, the pile in this model system behaved relatively rigid compared to that of other fixed model systems.

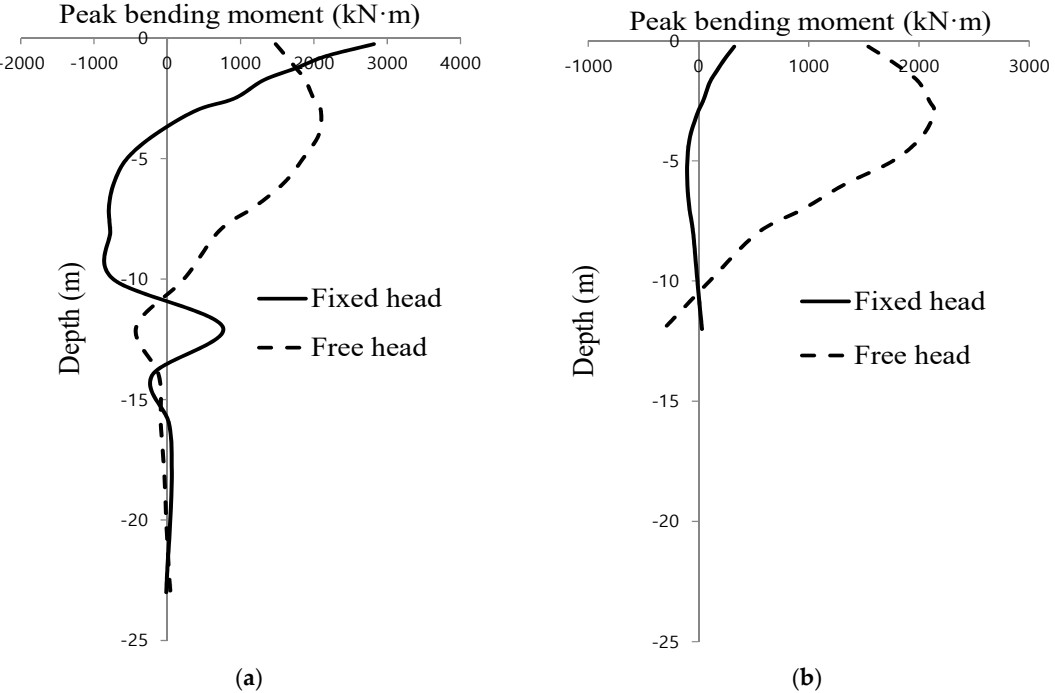

**Figure 21.** Peak bending moment profiles for different pile head fixity (input acceleration: 0.13 g). (**a**) Pile length: 23 m (12.6 T); (**b**) Pile length: 8 m (4.4 T).

## 5. Conclusions

A new 3D numerical modeling methodology is suggested to simulate the dynamic soil-pile-structure interaction in dry sand, in addition, the applicability of the proposed numerical model was verified by comparing with results of dynamic centrifuge tests. Parametric studies for various conditions were also performed to examine important characteristics of the dynamic pile behavior in dry sand. Detailed conclusions are as follows.

(1) The proposed numerical model employed non-linear elastic, Mohr–Coulomb plastic as a soil constitutive model with a hysteretic damping model to simulate nonlinear behavior and energy dissipation in soil media. The initial shear modulus, other various dynamic soil properties were determined by verified empirical relations and repetitive preliminary analysis. Simplified continuum modeling was adopted to simulate a semi-infinite boundary and to enhance analysis efficiency. Interface model was also properly applied to predict various interactions occurred at soil-pile boundary.

(2) Through a series of validation procedures, the applicability and viability of the proposed model were verified. The representative internal pile response profiles predicted by the proposed numerical model consistently show good agreement with those observed from the centrifuge model test both for various input earthquake conditions and pile sizes.

(3) From parametric study, the effect of the inertial force induced by the superstructure is dominant, whereas the effect of the kinematic force induced by soil movement is relatively not significant in dry sand. Pile length strongly affected not only in pile dynamic response trend according to the depth but also in rotational depth and occurrence depth of the peak bending moment. The pile head fixity was another important factor for dynamic soil-pile-structure interaction. It governed the peak bending-moment profile of pile and affected the dynamic responses of the system in conjunction with other factors, such as pile rigidity.

**Author Contributions:** S.Y.K. organized the paperwork, made a analysis plan, performed numerical analysis and suggested the numerical analysis method; M.Y. supported the numerical model verification by comparing test results helped the data analysis; all authors contributed to the writing of paper.

**Acknowledgments:** This research was supported by a grant from (1) R&D Program (PK1902A4) of the Korea Railroad Research Institute, Republic of Korea. (2) Review of Environmental Impact Assessment funded by Korea Environment Institute.

**Conflicts of Interest:** The authors declare no conflict of interest.

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
