# Peer review of "Evaluation of Dynamic Soil-Pile-Structure Interactive Behavior in Dry Sand by 3D Numerical Simulation"

_applsci, doi:10.3390/app9132612_

Round 1

Reviewer 1 Report

In this manuscript, authors have numerically studied the dynamic response of soil-pile interaction. The manuscript is well-prepared and can be considered for publication after carefully addressing the following comments: 

Some of the figures of the manuscript are similar to ones previously used and published by the authors (see Ref 1 below). The figures should be revisited or a copyright statement should be given in the figure caption. 

There are some papers when the analytical treatment of soil-structure interaction was analytically studied (see Ref 2-6 below). Expanding the introduction of the manuscript to address these papers would bring more audience to the current manuscript. 

There are some glitches in the equations and their definitions of the manuscript that should be fixed. For instance, "L_2" term after equation 1. 

Authors are recommended to use better quality images. Right now all the figures are getting rasterized. 

In Figure 6, proper labels should be used to separate subfigures. You may want to use "a" and "b". Also, more information should be given for the figure captions. 

References

1. "Numerical Simulation of Dynamic Soil-pile Interaction for Dry Condition Observed in Centrifuge Test." Journal of the Korean Geotechnical Society 32.4 (2016): 5-14.

2. "Vibration analysis of a rigid circular disk embedded in a transversely isotropic solid." Journal of Engineering Mechanics 140.7 (2013): 04014048.

3. "Lateral translation of an inextensible circular membrane embedded in a transversely isotropic half-space." European Journal of Mechanics-A/Solids 39 (2013): 134-143.

4. "Dynamic analysis of a rigid circular foundation on a transversely isotropic half-space under a buried inclined time-harmonic load." Soil Dynamics and Earthquake Engineering 63 (2014): 184-192.
5. “Forced vertical vibration of rigid discs with an arbitrary embedment.” Journal of Engineering Mechanics 117. 11 (1991): 2527– 2548
6. "Vertical vibrations of a rigid disk embedded in a poroelastic medium." International Journal for Numerical and Analytical Methods in Geomechanics 23.15 (1999): 2075-2095.

Author Response

Please, check the attached file.

Reviewer 2 Report

 The manuscript needs to be improved significantly with major modifications.

1)    Comment on manuscript  in general

The verification of the model   has been already  published before in Proceedings of the 18th International Conference on Soil Mechanics and Geotechnical Engineering, Paris 2013.

with following manuscript  

3D Dynamic Numerical Modeling for Soil-Pile-Structure Interaction in CentrifugeTests

The only contribution of the authors is the parametric studies on  the 3D model.

The authors  should clarify that clearly on the manuscript .

Author Response

Please, check the attached file.

Reviewer 3 Report

Thank you for this paper. 

Some comments: 

- Some editing issues remain, please read again and correct. 

- Lines 110 - 117: please explain the procedure to obtain the experimental value (A and n in particular). 

- It would have been interesting to compare these results, especially the parametric studies, with FEM results. Coudl you please consider to give some input on that?

- In the discussions/conclusion part, comment on the similarity reduced scale/full-scale structure,and the translation of the obtained results. 

Author Response

Please, check the attached file.

Round 2

Reviewer 1 Report

Authors have addressed my comments. I do recommend the manuscript for publication. 

Reviewer 2 Report

The authors have adequately addressed the comments raised in the previous review